

# Integrating Topographic Continuity and Lake Recession Dynamics for Improved Bathymetry Mapping from DEMs

Fukun Tao[1], Yong Wang[1], Yinghong Jing[1], Xiaojun She[1], Shanlong Lu[2], Yao Li[1]

[1] Chongqing Jinfo Mountain Karst Ecosystem National Observation and Research Station, School of Geographical Sciences,
Southwest University, Chongqing 400715, China

[2] Aerospace Information Research Institute, Chinese Academy of Sciences, Beijing 100094, China

*Correspondence to*: Yao Li (liyao7@swu.edu.cn)

**Abstract.** Accurate lake bathymetry is critical for advancing hydrological and biogeochemical research, yet large-scale and deep-water mapping remains constrained by cost challenges. While remote sensing techniques have been extensively
employed for bathymetry mapping, their effectiveness is primarily limited to shallow waters due to the rapid attenuation of optical signals with increasing depth. To overcome this limitation, we propose a novel bathymetry mapping method that leverages topographic continuity to infer underwater terrain by simulating lake level recession dynamics. This approach relies solely on Digital Elevation Model (DEM) data, using shoreline topographic gradients to estimate depth, providing a robust alternative for regions where conventional surveying is impractical. Validation across 12 lakes on the Tibetan Plateau
demonstrated promising accuracy, with an average normalized root mean square error of 19.08% for depth estimation and a mean absolute percentage error of 23.47% for lake volume. To evaluate the method's generalizability across diverse hydrological settings, it was applied to Lake Mead, United States, producing a bathymetry map with a correlation coefficient of 0.66 against in situ measurements. Overall, this study introduces a low-cost solution for bathymetry mapping in data-scarce regions, offering a valuable tool for assessing lake volume at regional and global scales.

**1 Introdution**

Lakes, covering approximately 2% of the global land surface, play a critical role in water resource assessments and water balance analyses (Råman Vinnå et al., 2021; Pekel et al., 2016). They act as essential mediators of water-energy exchanges and serve as sensitive indicators of climate change at both regional and global scales (Vinogradova et al., 2025). By directly influencing water resource equilibrium, lakes are integral to hydrological and ecological processes (Williamson et al., 2009).
Bathymetry data and lake volume —two key physical parameters of lakes—are fundamental for various applications, including hydrodynamic modeling (Yoon et al., 2012; Durand et al., 2008), aquatic vegetation monitoring (Gong et al., 2021), water resource management (Li et al., 2023; Yang et al., 2021), and climate model evaluation (Woolway et al., 2020). Therefore, accurate measurements of lake depth and volume are essential for environmental conservation and sustainable



resource management. However, the efficiency of current bathymetry mapping technologies remains limited, substantially
compromising the accuracy of lake volume estimates and restricting their practical applications. This underscores an urgent
need for methodological innovations to advance lake depth estimation and volume prediction.

Traditional in situ bathymetric surveys rely on shipborne echo sounders, airborne LiDAR, and optical imaging sensors (Guo
et al., 2022; Song et al., 2014; Smith and Sandwell, 1997). While these techniques provide high-resolution data, they are
costly and impractical for large-scale lake assessments. Satellite remote sensing offers a cost-effective alternative for global
inland water monitoring (Li et al., 2021b; Sheffield et al., 2018), yet accurately mapping underwater topography remains a
significant challenge. Optical remote sensing is widely used for bathymetry mapping in shallow waters (Wang et al., 2024),
but its effectiveness deteriorates exponentially in deeper lakes due to the limited penetration of light (Roy and Das, 2022).
Radar altimetry and Digital Elevation Models (DEMs) can capture topographic features but exhibit fundamental limitations
in characterizing submerged terrain. Notably, prevailing global DEM products assign a fixed elevation value to water-
covered areas, reflecting surface elevation rather than actual underwater topography (Li et al., 2021a; Yamazaki et al., 2017;
Farr et al., 2007). Therefore, these products depict submerged terrain only in areas that became inundated after the time of
observation, failing to deliver a comprehensive representation of lake bathymetry.

In recent years, the integration of satellite altimetry and optical remote sensing data has been increasingly employed to
monitor inland water depth (Li et al., 2019; Qiao et al., 2019; Duan and Bastiaanssen, 2013). Lake surface areas are typically
derived from optical imagery, while water surface elevations are obtained from satellite altimetry data (Li et al., 2021a). By
correlating the time series of the lake area with corresponding elevation measurements, area-elevation relationships can be
established (Luo et al., 2021; Gao, 2015). Based on these relationships, lake volume is estimated through geometric volume
calculation formulas tailored to specific lakes. For instance, Li et al. (2020) combined multi-source satellite altimetry data
with Landsat imagery to establish area-elevation relationships, which were subsequently utilized to estimate dynamic
reservoir depths. They further employed an extrapolation method to generate a complete bathymetric dataset. Although this
method enables comprehensive bathymetry mapping, it depends on prior knowledge of the maximum lake depth as a
constraint, limiting its applicability in regions where such information is unavailable.


Shoreline topography plays a crucial role in determining lake depth (Messager et al., 2016; Pistocchi and Pennington, 2006),
as the morphology and slope of the shoreline significantly impact water flow and sedimentation processes (Edmonds and
Slingerland, 2010). Therefore, lake shoreline characteristics provide valuable input for lake depth estimation. Some studies
have employed statistical models to analyze large datasets of known lake and reservoir depths. Nonlinear models based on
surrounding topography have been developed to estimate average or maximum water depth, which is then used to calculate
lake volume (Han et al., 2024; Cael et al., 2017; Messager et al., 2016). While such methods are generally reliable for large-
scale studies due to compensatory effects of under- and over-estimations, uncertainties and biases remain substantial at local



or individual lake scales. Alternative approaches suggest that lake bathymetry can be inferred by extrapolating or interpolating from the surrounding terrain (Liu and Song, 2022; Getirana et al., 2018). Although these approaches can

partially reconstruct underwater topography and estimate water volume, they often require prior in situ data, such as maximum lake depth, to constrain the results. Recently, several methods have been developed that eliminate the need for field measurements (Han et al., 2024; Fang et al., 2023; Zhu et al., 2019). Due to sediment accumulation, most modern lakes tend to develop deep-water zones with relatively flat bottoms (Zhang et al., 2018). Current DEM-based bathymetry mapping methods often truncate predictions when the estimated depth exceeds measured values (Zhang et al., 2016), leading to

artificially flattened lake bottoms. However, such truncation fails to account for sedimentation processes and the variations in underwater topography caused by sediment accumulation.

To address these limitations, this study proposes a novel three-dimensional bathymetry mapping approach that utilizes DEM data to estimate lake depth and volume. The method leverages shoreline geometric features and topographic factors extracted

from DEMs, while explicitly accounting for the physical processes of water level recession and sediment deposition. By capturing the reshaping effects of sediment accumulation on lakebed morphology, this approach provides a more accurate representation of underwater topography and supports improved lake volume estimates. Consequently, it offers a promising solution for generating bathymetry maps and estimating lake volume in natural lakes, even in regions with limited in situ data.

**2 Data and Methods**

**2.1 Data**

**2.1.1 Digital Elevation Model datasets**

NASADEM, an enhanced version of the Shuttle Radar Topography Mission (SRTM) DEM with a 1 arc-second (~30 m at the equator) spatial resolution, was used as the primary input dataset (Crippen et al., 2016). One of its key advantages for

bathymetry mapping is its early acquisition time, which captures more exposed shoreline areas, thereby facilitating more comprehensive lake depth estimations. It has been reprocessed using advanced algorithms to refine the original SRTM radar signal, incorporating precise elevation benchmarks primarily from the Ice, Cloud, and Land Elevation Satellite (ICESat) Geoscience Laser Altimeter System (GLAS) and the Advanced Spaceborne Thermal Emission and Reflection Radiometer (ASTER). Validation studies across the Tibetan Plateau have confirmed its high vertical accuracy, demonstrating a low root

mean square error (RMSE) of 3.36 m in the validation area (Li et al., 2022). This dataset is publicly available through NASA's Earthdata portal (https://www.earthdata.nasa.gov). To assess the impact of input data with different spatial resolutions on the simulation results, we incorporated two additional datasets: ALOS PALSAR DEM (12.5 m spatial resolution) and MERIT DEM (3 arc-second resolution, ~90 m at the equator). The ALOS PALSAR DEM data were obtained



from the Japan Aerospace Exploration Agency (JAXA, https://www.eorc.jaxa.jp), while the MERIT DEM data were sourced

from the International Centre for Water Hazard and Risk Management (ICHARM, https://hydro.iis.u-tokyo.ac.jp).

### 2.1.2 In situ bathymetry data

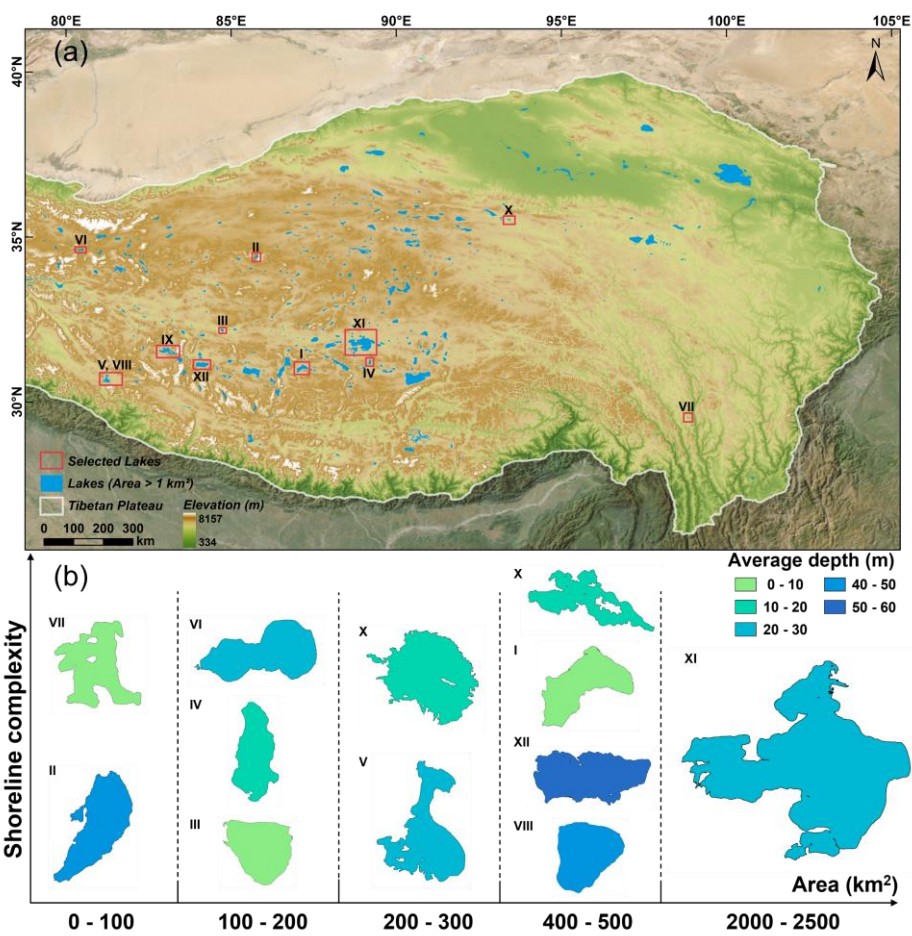

**Figure 1: Overview of the 12 sample lakes on the Tibetan Plateau. (a) Distribution of lakes larger than 1 km² across the Tibetan**
**Plateau, with the sample lakes in this study highlighted by red rectangles. (b) Shape characteristics of the 12 sample lakes, where**
**different colors represent variations in mean lake depth.**

The in situ data were provided by the National Tibetan Plateau/Third Pole Environment Data Center (http://data.tpdc.ac.cn),

which were collected using an echo sounder (Lowrance HDS5). Spatial interpolation was performed at a 30 m resolution,

utilizing bathymetric points containing longitude, latitude, and depth values to generate in situ bathymetry maps. The

shoreline shapefiles were extracted at the zero-depth boundary (depth = 0 m) and used to calculate lake surface areas. The

selected lakes, which are evenly distributed across the Tibetan Plateau, exhibit diverse morphologies. Their mean water



surface elevations range from 4,299 m to 5,171 m, spanning an altitude difference of approximately 1,000 m. Most lakes have surface areas between 90 and 500 km², except for Siling Co, which is significantly larger at approximately 2,400 km².

Maximum lake depths range from 4 m to 130 m, reflecting diverse bathymetric features and shoreline complexities. Detailed lake characteristics are presented in Table 1. Additionally, to assess the method's applicability to disparate hydrological contexts, a 30 m resolution bathymetry map of Lake Mead—an artificial reservoir—was obtained from the United States Geological Survey (USGS, https://www.usgs.gov/). This dataset includes information on post-impoundment sediment distribution within the lake. However, due to the limited coverage of the in situ bathymetry map, portions of the eastern

region of Lake Mead were excluded from the analysis.

**Table 1.** Overview of the sample lakes on the Tibetan Plateau.

| Lake ID | Lake name | Area (km²) | Average elevation (m) | Depth (m) Average | Depth (m) Maximum |
|---|---|---|---|---|---|
| I | Angzicuo | 495.42 | 4693 | 9.70 | 18.83 |
| II | Buruocuo | 92.67 | 5171 | 41.63 | 100.55 |
| III | Dongcuo | 106.79 | 4397 | 2.06 | 3.99 |
| IV | Guomangcuo | 113.63 | 4634 | 15.48 | 39.49 |
| V | Laangcuo | 252.56 | 4571 | 21.90 | 49.19 |
| VI | Longmucuo | 106.80 | 5009 | 25.27 | 67.52 |
| VII | Mangcuo | 19.39 | 4299 | 9.41 | 22.28 |
| VIII | Mapang Yongcuo | 413.22 | 4585 | 41.62 | 79.45 |
| IX | Ngangla Ringco | 498.06 | 4715 | 18.26 | 74.94 |
| X | Salt Lake | 209.90 | 4469 | 13.13 | 32.78 |
| XI | Siling Co | 2389.11 | 4539 | 22.07 | 52.50 |
| XII | Taro Co | 487.49 | 4570 | 57.48 | 130.95 |

### 2.1.3 Comparative datasets

The estimated lake volumes were compared with those reported in three previous studies. GLOBathy is a recently developed global dataset that generates bathymetry maps based on maximum depth estimates, combined with the geometric and physical properties of water bodies from the HydroLAKES dataset (Khazaei et al., 2022). Han et al. (2024) adopted an improved grid-based photon noise removal method to extract underwater photon signals from ICESat-2 data. By integrating statistical modeling with self-affine theory, they generated lake bathymetry maps and estimated lake volume for lakes larger





than 0.01 km² on the Tibetan Plateau. Fang et al. (2023) employed a trigonometric function as the fundamental shape of depth profiles and applied error correction based on different water level elevations to reduce the impact of sediment layers on simulation accuracy. The method was applied to 12 lakes on the Tibetan Plateau, and the results were included in the comparison.

## 2.2 Method

The bathymetry mapping algorithm consists of three main steps (Fig. 2). Parameter initialization: Key input parameters for underwater elevation estimation, including the water mask and shore slope, were extracted from the updated SRTM Water Body Data (SWBD) and elevation bands of NASADEM. The water mask defined the spatial extent of the computation, while the shore slope served as the independent variable for estimating underwater elevation. Underwater elevation calculation: The water mask was treated as the initial lake surface. A simulated recession process was then applied,

progressively lowering the water level and reducing the water mask coverage. This simulation process, initially proposed by Zhu et al. (2019), has been applied in recent studies. However, previous studies required manual predefinition of the water level drop at each step. In this study, the process was improved to automatically simulate the entire recession without manual intervention. A detailed description of the simulation process is provided in Section 2.2.2. At each step of the simulated retreat, elevation values for newly exposed underwater pixels were calculated. This iterative process continued until all

pixels within the original water mask were assigned updated elevation values. Consequently, the original water surface elevations in the DEM were progressively replaced with estimated underwater elevations. Lake volume estimation: The updated DEM, containing elevation values for underwater pixels, was combined with the lake extent to generate a complete bathymetry map. From this map, lake depth and volume were calculated. This structured approach ensured accurate bathymetry mapping by iteratively estimating underwater elevations based on shoreline topography and water level recession.


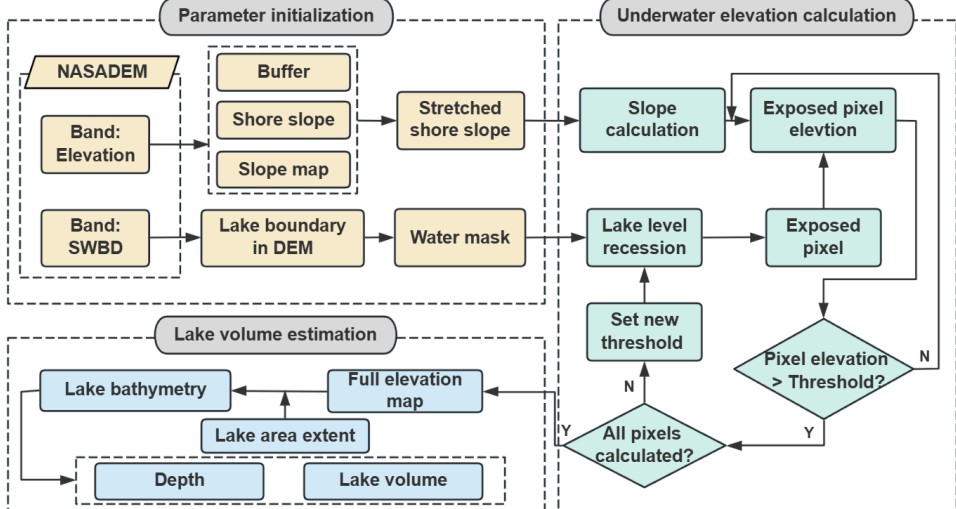

**Figure 2: Flowchart of bathymetry generation and lake volume estimation.**



### 2.2.1 Parameter initialization

The DEM dataset used in this study assigned the lake surface elevation to all water-covered pixels. To distinguish water bodies from surrounding land areas, a binary land-water classification was generated by filtering the SWBD band values in the DEM, where pixels with a value of 255 denoted water. As the initial classification may contain fragmented water patches, a connected component detection algorithm was applied to eliminate these artifacts. Only the largest connected region was considered the main lake body. Once the main lake body was identified, an edge detection algorithm was employed to

extract the shoreline pixels. A new matrix—referred to as the water mask matrix—was then generated with the same spatial reference as the input DEM (Fig. 3b).

A multi-level buffering strategy was employed to define the spatial extent of the DEM used in subsequent calculations (Fig. 3a). A maximum buffer zone of 600 m (approximately 15–20 pixels) was established around the lake boundary in DEM (LBD). Within this zone, nested buffer zones were generated at 100 m intervals (about 3–4 pixels). The average elevation

within each buffer was calculated, producing a series of elevation values corresponding to increasing distances from the LBD.

A terrain transition point was identified by progressively sampling outward from the LBD to determine where the elevation trend shifted from an increase to a decrease. The edge at this transition point was defined as the outer edge of the final buffer

extent, thereby ensuring that only terrain relevant to shoreline slope estimation was included. If no transition point was detected, the default buffer of 600 m was used.

Elevation values within the final buffer zone were used to estimate the initial shore slope. As illustrated in Fig. 3b, for each

lake boundary pixel in the water mask matrix, an 8-neighborhood detection method was applied to identify adjacent land pixels in the landward direction. Using the DEM's elevation band, elevation values were extracted along this landward direction within the buffer zone (Fig. 3c). A linear or polynomial regression was then applied to estimate the slope. Once all landward slopes were computed, their average value was used as the initial shore slope, which served as a key input for underwater elevation estimation. This process was repeated for all LBD pixels.






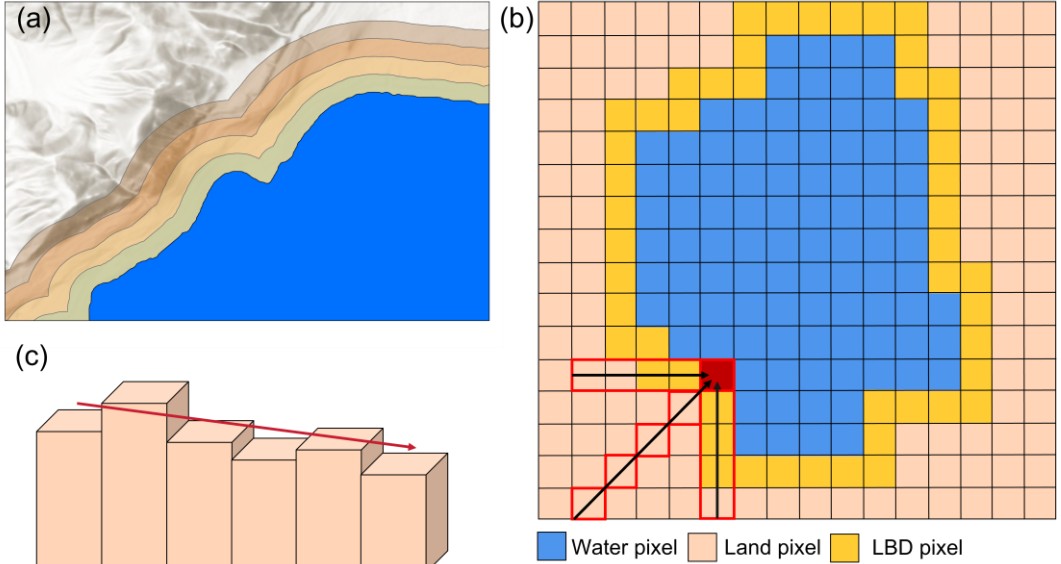

Water pixel  Land pixel  LBD pixel

**Figure 3: Shoreline slope estimation process for LBD. (a) Multi-level buffer zones surrounding the lake boundary. (b) Slope calculation using linear regression within the buffer zone. The red pixel indicates the target shoreline pixel under calculation, while the red rectangular box represents the surrounding pixels involved in the calculation. (c) Linear fitting process in a specific direction, with the red line representing the fitted slope result.**

While the computed shore slope accounts for local terrain orientation, relying solely on a linear regression within a fixed window may not fully capture the elevation variability within the buffer zone. To better represent the general terrain transition from the shoreline toward the lake center, the method integrated the directional shore slope with broader topographic variation observed across the entire buffer zone. A slope map was generated for the buffer zone by applying a 3×3 moving window to estimate the slope at each pixel. This approach captured localized terrain undulations and provided a more comprehensive representation of elevation variation within the buffer. To enhance statistical robustness and reduce the influence of extreme values, outliers in the slope data were removed using a 95th-percentile threshold. The previously computed directional shore slope values were then rescaled to match the value range of the slope map using the following transformation formula:

$$S_1'(i,j) = \min(S_2) + \frac{S_1(i,j) - \min(S_1)}{\max(S_1) - \min(S_1)} \times (\max(S_2) - \min(S_2)) , \tag{1}$$

### 2.2.2 Underwater elevation calculation

Based on the principle of terrain continuity, underwater areas closer to the shoreline tend to exhibit greater similarity to the topographic characteristics of the shoreline. Following this principle, the elevation values of underwater pixels near the shoreline were iteratively estimated using the following formula:



$$h_i = h_i' - d \cdot k_i, \tag{2}$$

where $h_i$ represents the elevation of the $i$-th pixel, $h_i'$ is the elevation of the adjacent land pixel, d is the distance between the two pixels, and $k_i$ is the slope at the location of the $i$-th pixel. As shown in Eq. (2), calculating the elevation of the $i$-th pixel requires the known elevation value of a neighboring pixel (i.e., $h_i'$) as input. Therefore, the sequence and pattern in which underwater elevations were calculated significantly influenced the final bathymetric results. This approach simulated the physical process of water level recession, gradually exposing underwater pixels. A loop-based method was then employed to iteratively calculate the elevation of these pixels following topographic gradients.

As illustrated in Fig. 4a, this process improved the simulation of lake recession. During each iteration, the lake level decreased incrementally, exposing a new set of underwater pixels. In steep regions, fewer underwater pixels were exposed per iteration due to rapid elevation changes, while in flatter areas, more pixels became exposed at once. As each pixel was exposed, its elevation was calculated sequentially, preserving spatial continuity and topographic consistency.

The first iteration began at the boundary of the water mask, where the elevations of all pixels adjacent to the shoreline were calculated. Among these newly computed pixels, the minimum elevation was identified and set as the threshold for the initial water level drop. Once the elevation of an underwater pixel was determined, its status in the water mask was updated from water to land, indicating the transition from submerged to exposed. Following this, a new inner boundary was defined, and its elevation values were calculated. Pixels with elevation values greater than or equal to the updated threshold were retained for the next round of calculations. This iterative process continued until, within a given iteration, all calculated elevations were less than or equal to the threshold. At that point, the loop ended, and a new iteration began with an updated threshold. With each iteration, the shoreline elevations became progressively more uniform, simulating a realistic recession of the water surface. The process repeated until no water pixels remained in the water mask, at which point the entire lake bottom was assigned elevation values, and the bathymetric computation was complete.

It is essential to recognize that the underwater topography of a lake cannot be simplified as a single basin, as multiple sub-basins may exist Fig. 4b. Therefore, in this study, no constraints were imposed during the simulation of lake recession. Once newly exposed land pixels connected to the shoreline, the lake was segmented into multiple sub-basins. The recession process was then applied to each sub-basin individually through boundary detection. This approach avoids the oversimplified assumption that all lakes consist of a single basin, instead accurately capturing the true complexity of the lakebed morphology.




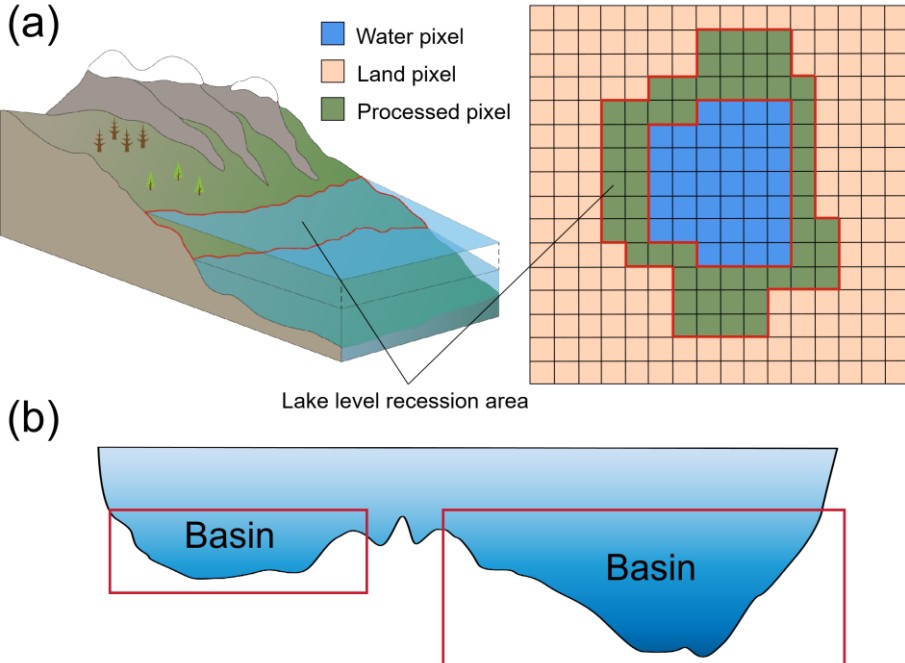

**Figure 4: Simulation of the physical process of lake level recession. (a) Simulation of the physical process of lake level recession during bathymetric calculation. The red line represents the lake boundary before and after each water level drop. (b) The profile of a lake along its central axis. The red rectangle highlights the extent of the sub-basin within the lake.**

During the calculation process, most water pixels were adjacent to multiple land pixels, resulting in several potential elevation estimates for each water pixel. To determine the most reliable elevation, a weighted average operation was applied to the estimates from each direction. The weight for each direction was based on the distance from the water pixel to the opposite shore along that direction. The specific calculation formula is as follows:

$$\begin{cases} H = \sum_{i=1}^{n} H_i \cdot w_i \\ w_i = \frac{1/d_i}{\sum_{j=1}^{n} 1/d_j} \end{cases}, \tag{3}$$

where $H$ represents the final calculated elevation of the water pixel, $H_i$ represents the elevation estimate from direction $i$, $w_i$ represents the weight assigned to direction $i$, $d_i$ represents the distance from the water pixel to the opposite shore in direction $i$, and $n$ represents the total number of directions with adjacent land pixels, $d_j$ represents the distance from a water pixel to the opposite shore in direction $j$, used for normalization. The final elevation of each water pixel was determined as a weighted average of estimates from all valid directions. Under the assumption that pixels closer to the shoreline provide more reliable topographic information, elevation estimates associated with shorter distances to the shore were assigned higher weights.




In Eq. (2), the parameter $k_i$ plays a crucial role in modeling underwater topography at each pixel. It reflects the lake's cross-sectional shape, and different assumptions about this shape can significantly affect the simulation outcomes (Liu et al., 2020; Messager et al., 2016). For simplification, the method assumed that the lowest point along the lake's vertical profile could serve as the origin of a local coordinate system, as illustrated in Fig. 5. This lowest point typically lay closer to the steeper shore. To determine its position, shore slopes on both sides of the profile were first extracted. The assumed location of the

lowest point was then calculated based on the relative slope values of the two shores. Taking the left shore as an example, the horizontal distance between the lowest point and the left shoreline was determined by:

$$L = D_i \cdot \frac{s_r}{s_l + s_r}, \tag{4}$$

where $D_i$ is the total horizontal distance between the two shores along the current profile. $s_l$ and $s_r$ represent the slope values of the left and right shores, respectively. After determining the horizontal position of the lowest point, the model assumed

that the slope at this point asymptotically approached zero, representing a local minimum in the profile. Under this assumption, the slope at the $i$-th pixel along the profile is assumed to decrease linearly with increasing distance from the shoreline. The slope variation is defined by the following formula:

$$k_i = s_l \cdot (1 - \frac{l}{L}), \tag{5}$$

where $l$ represents the horizontal distance from the $i$-th pixel to the left shore along the profile. Although Eq. (5) provides an

initial estimate of the underwater topographic slope along the profile, continuous sedimentation gradually reshapes the lakebed, leading to a progressively flatter underwater terrain.

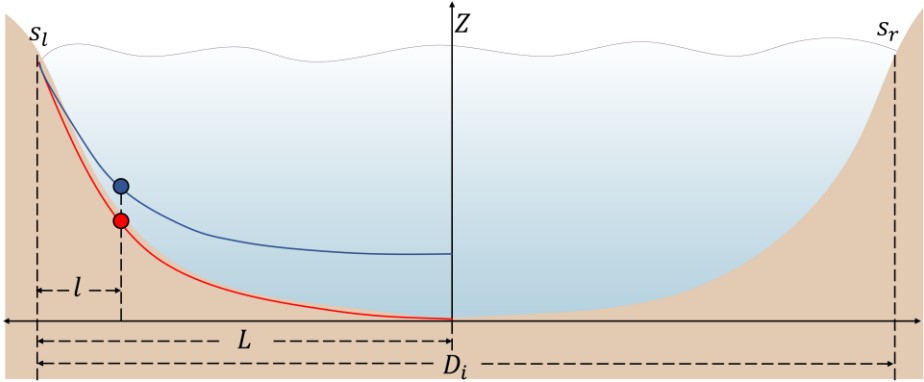

**Figure 5: Schematic diagram of the lake profile coordinate system. The red line represents the lakebed profile before applying Eq.**

**(6), while the blue line shows the profile after correction. The red and blue points indicate the positions of the calculation points along the profile before and after the correction, respectively.**





To further enhance the simulation accuracy, a correction was introduced to account for sediment accumulation and its impact on underwater topography. It is assumed that sediment tends to accumulate more heavily with increasing distance from the
shoreline, leading to a progressively flatter lake bottom (Håkanson, 1982). Incorporating this effect was expected to improve the realism of the simulation by aligning the modeled terrain with observed sedimentation patterns. The correction was applied to the initially estimated slope using the following formula:

$$k_i' = k_i \times \left(\frac{d_i}{D_i}\right)^\alpha, \tag{6}$$

where $k_i'$ is the corrected slope at the $i$-th pixel, and $k_i$ is the uncorrected slope. $d_i$ refers to the distance from the $i$-th pixel
on the current land boundary to the opposite shore along the profile. The exponent $\alpha$ reflects the influence of the surrounding terrain on sediment accumulation. To determine a representative value for $\alpha$, we analyzed shoreline slope data from over 4,000 lakes on the Tibetan Plateau using the HydroLAKES dataset. The resulting distribution (Fig. S1) indicated an average shoreline slope of approximately 5.14°, which was rounded to 5° as a reference threshold for assigning the value of α. The underlying assumption is that steeper shorelines facilitate greater sediment transport toward the lake center, resulting in the
formation of flat sediment layers on the lakebed (Ju et al., 2012). Accordingly, to avoid overestimating the maximum depth in steep-shoreline lakes, $\alpha$ is determined as follows:

$$\begin{cases} \alpha = 1, \ 0° < \bar{\theta} \leq 5° \\ \alpha = 2, \ 5° < \bar{\theta} \end{cases}, \tag{7}$$

where $\bar{\theta}$ is the average slope within the defined buffer area, which was extracted during the "Parameter initialization" step.

### 2.2.3 Lake volume estimation

Following the underwater elevation calculation steps, a DEM representing the lake's underwater topography was generated within the extent defined by the water mask. It is important to note that this water mask was derived from the original DEM data and therefore reflected the lake boundary at the time the DEM was generated. To validate the accuracy of the simulated lake volume results, we used the boundary (where depth is 0) from the measured data as the lake extent. The observed boundary was overlaid onto the simulated elevation map to generate a lake depth map, from which lake depth and volume
values were extracted. The formula used to estimate lake volume is as follows:

$$V = \sum v_i = \sum s_i \times depth_i, \tag{8}$$

where $V$ is the volume of the lake, $v_i$ is the volume of the $i$-th water pixel, $s_i$ is the area of the $i$-th pixel (equal to the product of the pixel's length and width), and $depth_i$ is the water depth at the $i$-th pixel.

Since in situ bathymetry data did not include the water surface elevation at the time of measurement, a simple method was employed to extract this elevation. DEM elevation values along the observed lake boundary were extracted, and outliers





were removed. The water surface elevation was then estimated by fitting the distribution of these elevation values to a Generalized Extreme Value (GEV) distribution (Tseng et al., 2016; Morrison and Smith, 2002). Once the water surface elevation was obtained, the bathymetry map was converted to elevation values by subtracting the water depth from the water

surface elevation.

### 2.2.4 Accuracy evaluation

To assess the spatial differences between the simulated and measured water depths, 2,500 validation points were randomly selected across the surface of each lake. A linear regression analysis using the least squares method was conducted to examine the relationship between the simulated depths and in situ measurements. The evaluation metrics included the

Pearson correlation coefficient ($r$), normalized root mean square error (NRMSE), and mean absolute error (MAE). In addition, the difference between estimated and measured lake volume was quantified using the percentage error (PE).

### 3 Results

### 3.1 Bathymetry mapping and lake volume estimates

Three-dimensional visualizations of the simulated and in situ bathymetry maps (Fig. 6) were generated to compare their

spatial patterns. The results demonstrated that the simulated bathymetry closely aligned with the in situ water depth measurements, accurately capturing variations in underwater terrain. Among all the sample lakes, Dongcuo (Fig. 6c) exhibited the largest spatial discrepancy, likely due to its relatively flat shoreline, which limited the ability to infer underwater topographic patterns from the shoreline terrain. In contrast, Ngangla Ringco (Fig. 6i), with a more complex and fragmented shoreline, introduced localized inconsistencies in the simulated topography. As a result, the model tended to

overestimate depths near the shore. Furthermore, as many salt lakes have expanded significantly in recent years, a noticeable discrepancy was observed between the in situ shoreline and the boundary extracted from the DEM. Therefore, the bathymetry near the shoreline—derived from the DEM in the transitional area between these boundaries—was observed to be less smooth than the in situ bathymetry.





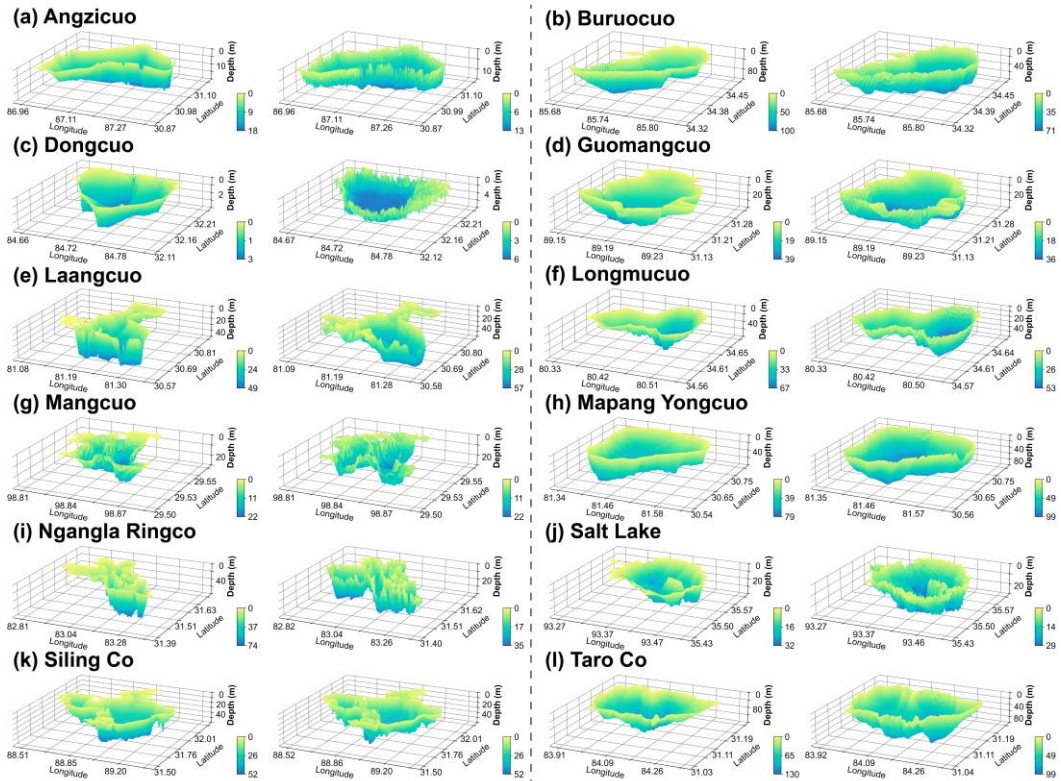

**Figure 6: Three-dimensional bathymetry maps of the sample lakes, derived from in situ measurements (columns 1 and 3) and corresponding simulated results (columns 2 and 4).**

To quantitatively evaluate the accuracy of the simulated bathymetry, 2,500 validation points were randomly generated within the boundary of each lake and compared against in situ depth measurements. As shown in Fig. 7, the simulated depths exhibited good agreement with in situ depths, with an average r value of 0.72 and an average NRMSE value of 19.09%. According to Table 2, the method underestimated the maximum water depth in several lakes, such as Angzicuo, Buruocuo, Longmucuo, Ngangla Ringco, and Taro Co. Among these lakes, Ngangla Ringco (Fig. 7i) showed the lowest level of agreement. A closer inspection of the three-dimensional bathymetry (Fig. 6i) revealed an abrupt increase in depth in the southern part of Ngangla Ringco. This localized variation posed challenges for the method, which relies primarily on shoreline topography to estimate underwater terrain. In contrast, the simulated water depths in Dongcuo and Mapang Yongcuo were overestimated. In particular, Dongcuo, with a surface area of 106.80 km² and a maximum water depth of only 3.99 m, demonstrated relatively poor simulation performance. Notably, MAE showed a strong correlation with lake depth ($r$ = 0.94). In deep lakes such as Buruocuo and Taro Co, where maximum water depths exceeded 100 m, MAE values reached 17.33 m and 21.15 m, respectively. These findings underscored that the correlation between underwater terrain and shoreline features tended to weaken with increasing depth. Nevertheless, the model maintained acceptable accuracy across a wide



range of lake sizes, depths, and morphologies, demonstrating its general applicability for regional-scale bathymetric estimation.

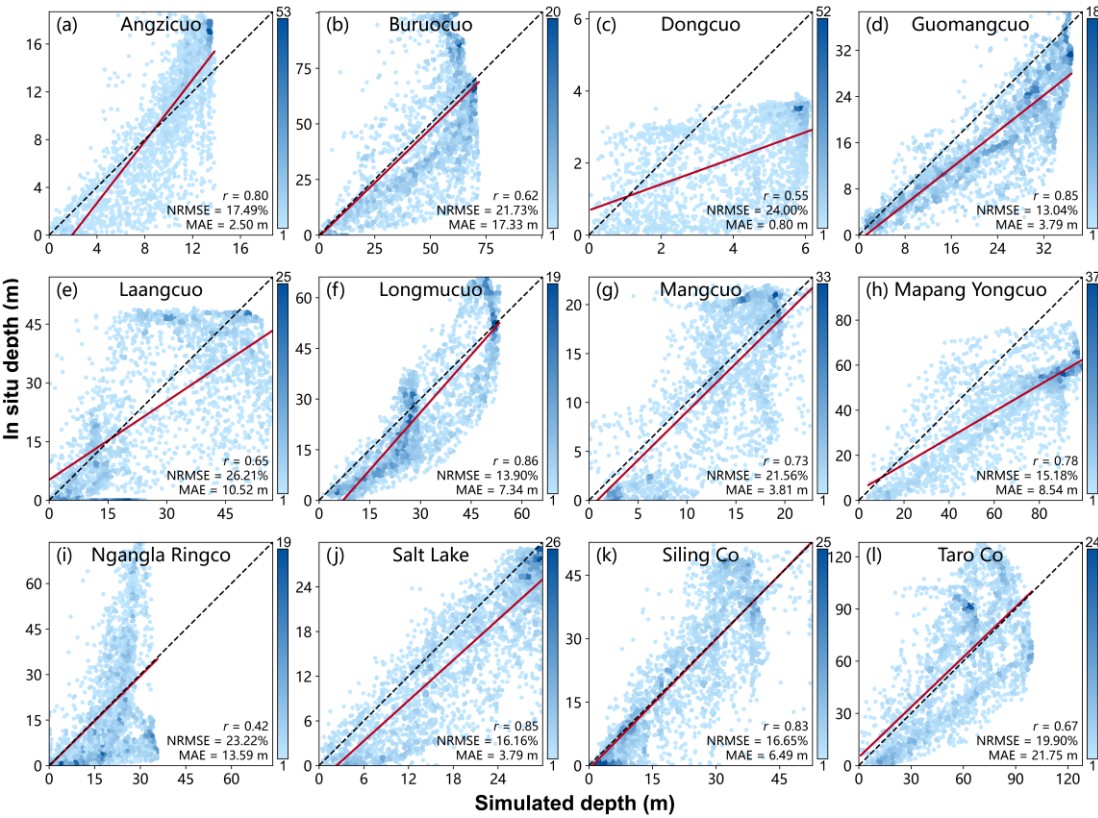

340

**Figure 7: Scatter plots comparing simulated lake depths with in situ measurements. The dashed line represents the 1:1 line, and the red line represents the linear regression fit.**





**Table 2.** Comparison of maximum depth and lake volume derived from in situ measurements and simulated bathymetry maps.

| Lake name | Maximum depth (m) | | PE | Volume (km³) | | PE |
|---|---|---|---|---|---|---|
| | In situ | Simulated | | In situ | Simulated | |
| Angzicuo | 18.83 | 13.85 | -26.46% | 4.81 | 4.65 | -3.17% |
| Buruocuo | 100.55 | 71.87 | -28.52% | 3.86 | 4.08 | 5.75% |
| Dongcuo | 3.99 | 6.36 | 59.37% | 0.22 | 0.43 | 94.84% |
| Guomangcuo | 39.49 | 36.83 | -6.73% | 1.76 | 2.38 | 35.14% |
| Laangcuo | 49.2 | 57.04 | 15.94% | 5.53 | 6.41 | 15.81% |
| Longmucuo | 67.52 | 53.45 | -20.84% | 2.70 | 3.08 | 14.12% |
| Mangcuo | 22.28 | 22.82 | 2.44% | 0.18 | 0.21 | 16.29% |
| Mapang Yongcuo | 79.45 | 99.17 | 24.82% | 17.20 | 26.70 | 55.22% |
| Ngangla Ringco | 74.94 | 35.70 | -52.37% | 9.09 | 9.57 | 5.30% |
| Salt Lake | 32.78 | 29.99 | -8.53% | 2.76 | 3.58 | 29.92% |
| Siling Co | 52.50 | 53.00 | 0.96% | 52.72 | 54.06 | 2.54% |
| Taro Co | 130.95 | 99.41 | -24.09% | 28.02 | 27.01 | -3.59% |

Lake volume was calculated from the simulated bathymetry maps to further assess the effectiveness of the proposed method. For most lakes, the method yielded satisfactory volume estimates (Table 2). The PE between simulated and in situ bathymetry maps for Angzicuo, Buruocuo, Ngangla Ringco, Siling Co, and Taro Co remained within 10%. Notably, despite a substantial underestimation of maximum depth in Ngangla Ringco, its overall lake volume estimate was still accurate. This result can be attributed to the offsetting effects between localized overestimations and underestimations in the simulated depth distribution. In contrast, Dongcuo exhibited a large discrepancy in lake volume estimation, with a PE of 94.84% compared to the volume derived from in situ bathymetry. This high error likely stemmed from the lake's shallow depth and small volume, making it highly sensitive to even minor elevation errors. Despite the underestimation of maximum depth in Angzicuo, Buruocuo, Longmucuo, and Taro Co, the volume estimates for these lakes remained reliable. This was because the overall depth distributions across the lake surface closely matched the in situ data, as illustrated in Fig. 7. In summary, among the 12 sample lakes, 10 exhibited volume estimation errors below 30%, with eight below 20%, demonstrating that the proposed method performed well in most cases.

**3.2 Validation of underwater terrain along transects**

To assess the effectiveness of the proposed method for simulating underwater topography, Fig. 8 compares simulated and measured depth profiles along designated transects for 12 lakes (transect locations shown in Fig. S2). Each transect was



carefully selected to align closely with the original depth survey routes while remaining near the lake center to ensure representative and consistent validation. The average NRMSE across the 12 transects was 27.34%, and the average MAE was 10.94 m, indicating an overall acceptable level of accuracy. In general, the simulated profiles were highly consistent

with in situ measurements, effectively capturing the overall trends in depth variation. When transects extended toward lake shores, the simulated elevation gradients matched the in situ measurements well, demonstrating the method's capacity to represent shoreline-to-bottom topographic transitions accurately. However, deviations were observed near the lake bottoms in several cases. Simulated depths were overestimated in Angzicuo (Fig. 8a), Buruocuo (Fig. 8b), Mangcuo (Fig. 8g), and Taro Co (Fig. 8l), while underestimations occurred in Dongcuo (Fig. 8c) and Mapang Yongcuo (Fig. 8h). In other cases—

such as Longmucuo (Fig. 8f), Ngangla Ringco (Fig. 8i), and Salt Lake (Fig. 8j)—although the general profile trends aligned well with observations, the method struggled to capture localized anomalies on the lake bottom. These discrepancies are likely linked to the complex geomorphological evolution of lakes on the Tibetan Plateau, which has experienced expansion, contraction, and migration processes over time. Such dynamic processes have resulted in heterogeneous sediment deposition and irregular bathymetric features (Yu et al., 2019). Consequently, a single correction formula may not fully account for

diverse lake morphologies. Despite these challenges, the proposed method demonstrated robust performance in simulating lake depth and bathymetry, supporting its applicability across varied lake environments.

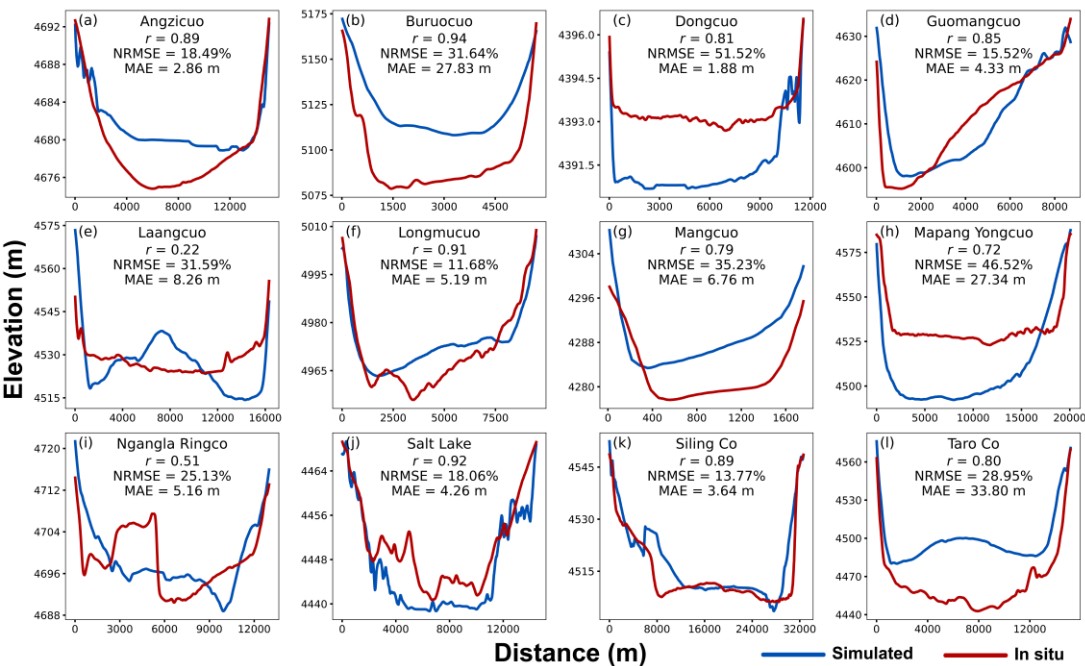

**Figure 8: Elevation profiles along validation transects for each lake. Each transect was selected to closely follow the corresponding**
**in situ survey route, providing representative comparisons between simulated and measured depths.**





## 4. Discussion

### 4.1 Comparison with previous studies

The simulated lake volumes for the 12 sample lakes were compared with results from three previous studies (Table 3). For
the GLOBathy dataset, absolute percentage error (APE) of lake volume estimates ranged from 1.79% to 324.93%, indicating considerable variability, with nearly half of the lakes exhibiting errors greater than 50%. In contrast, the dataset generated by Han et al. (2024) had an APE range of 0.52% to 375.20%, with four lakes showing APEs below 30%. The large errors in GLOBathy are primarily attributed to limitations in its bathymetry reconstruction method and uncertainties in estimating maximum depth. Although Han et al. (2024) proposed an ICESat-2-based bathymetry reconstruction method, their dataset
was derived from empirical equations. When applied to individual lakes, these equations can introduce unpredictable errors, which can account for the observed inconsistencies. Moreover, among the nine overlapping sample lakes reported in Fang et al. (2023), the average APE was 31.80%, outperforming both GLOBathy and Han et al. (2024). In comparison, our method achieved an average APE of 27.14% across the same nine lakes. Furthermore, among all 12 sample lakes, only Dongcuo, Guomangcuo, and Mapang Yongcuo exhibited PEs greater than 30%. These findings suggest that our method provides more
accurate and robust lake volume estimates compared to existing datasets.

In the comparative analysis, as the water storage estimates from Han et al. (2024) were available only through 2022, the corresponding water surface extent and water level elevation for that year were extracted, and the in situ water storage data were aligned to the same period for consistency. Since the water surface extent in GLOBathy's bathymetric maps is derived
from HydroLAKES, the lake volume estimates from GLOBathy were also adjusted to reflect the conditions in 2022. It should be noted that the boundary defined by the measured bathymetric data (depth = 0 m) was considered the lake shoreline in this study. Due to limitations in the bathymetric survey equipment, measurement uncertainties may exist within the recorded depth range.




**Table 3.** Comparison of percentage errors (PEs) in lake volume estimates for the sample lakes across different studies.

| Lake name | This study | Khazaei et al. (2022) | Han et al. (2024) | Fang et al. (2023) |
|---|---|---|---|---|
| Angzicuo | -3.17% | 80.28% | 375.20% | -24.46% |
| Buruocuo | 5.75% | -25.52% | -82.04% | 18.84% |
| Dongcuo | 94.84% | 324.93% | 339.99% | 63.88% |
| Guomangcuo | 35.14% | -21.07% | -40.01% | 19.47% |
| Laangcuo | 15.81% | 108.96% | 29.12% | - |
| Longmucuo | 14.12% | -47.18% | 35.44% | -51.83% |
| Mangcuo | 16.29% | 1.79% | -100.00% | - |
| Mapang Yongcuo | 55.22% | 69.19% | 0.52% | -36.87% |
| Ngangla Ringco | 5.30% | -70.62% | 20.98% | - |
| Salt Lake | 29.92% | -9.73% | 38.75% | 1.35% |
| Siling Co | 2.54% | 15.38% | 50.05% | 21.36% |
| Taro Co | -3.59% | -34.50% | 2.63% | -48.16% |
| Mean APE* | 23.47% | 67.70% | 92.90% | 31.80% |

* APE represents the absolute percentage error.

## 4.2 Impact of spatial resolution in DEM

We evaluated the effects of different input DEM datasets on water depth simulation and found that the resolution and quality of the input data significantly affect the simulation errors (Fig. 9). Overall, simulations based on NASADEM achieved the lowest mean APE, followed by MERIT DEM and ALOS PALSAR. The detailed evaluation results for ALOS PALSAR and MERIT DEM are presented in Tables S1 and S2, respectively. In addition, the error distributions for NASADEM and MERIT DEM were more compact, indicating greater stability compared to the more dispersed errors observed with ALOS PALSAR-derived results. While increasing DEM resolution from 90 m to 30 m led to a clear improvement in simulation accuracy, further refinement to 12.5 m using ALOS PALSAR unexpectedly resulted in decreased accuracy. Although recent studies have demonstrated the high vertical accuracy of ALOS PALSAR over the Tibetan Plateau (Xu et al., 2024), its performance in water depth simulation within our methodological framework was inferior to that of NASADEM. This discrepancy may be attributed to a combination of the intrinsic data quality and the sensitivity of the simulation algorithm to varying spatial resolutions, which together could explain the suboptimal performance of ALOS PALSAR in this context.



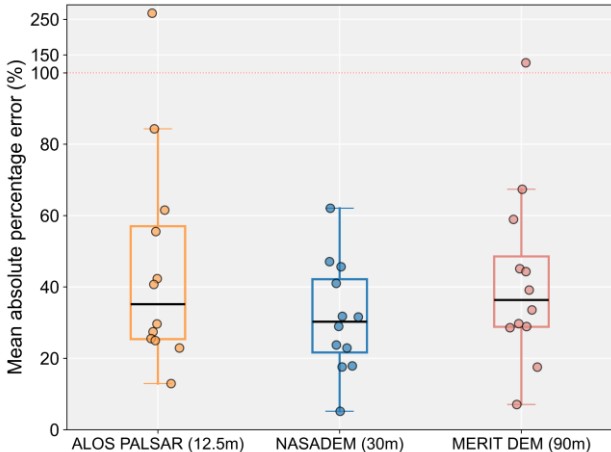

**Figure 9: Box plots showing the distribution of mean absolute percentage error in lake depth simulations using three digital elevation models (DEMs) with varying spatial resolutions: ALOS PALSAR (12.5 m), NASADEM (30 m), and MERIT DEM (90 m). Each dot represents the mean absolute percentage error for an individual lake. Box ranges represent the upper and lower quartiles, and whiskers extend to 1.5 times the interquartile range. The original bathymetric survey points were selected as sample points for water depth. The data used for this figure are provided in Table S3.**

We selected two moderately sized lakes (approximately 100 km²) with reliable simulation performance to examine spatial variations in depth estimates using different input datasets. Overall, the bathymetry maps generated at 12.5 m (Fig. 10b, f) and 30 m resolutions (Fig. 10c, g) exhibited broadly consistent spatial patterns. In contrast, the simulated bathymetry based on the 90 m resolution DEM (Fig. 10d, h) showed a pronounced underestimation of depth across the entire domain, consistent with the quantitative depth and volume results (Table 4). This bias arises because the proposed method operates at the pixel level, and coarser resolutions smooth the shoreline elevation, leading to an underestimated slope factor during the "parameter initialization" step. Consequently, coarser-resolution inputs produce gentler underwater topography, but this results in substantial depth underestimation in deep-water areas compared with in situ measurements. Increasing the resolution from 90 m to 30 m markedly improves the representation of deep-water areas, while further refinement to 12.5 m yields no notable change in spatial patterns. Therefore, considering both computational cost and accuracy, a 30 m input resolution is optimal for the proposed method.





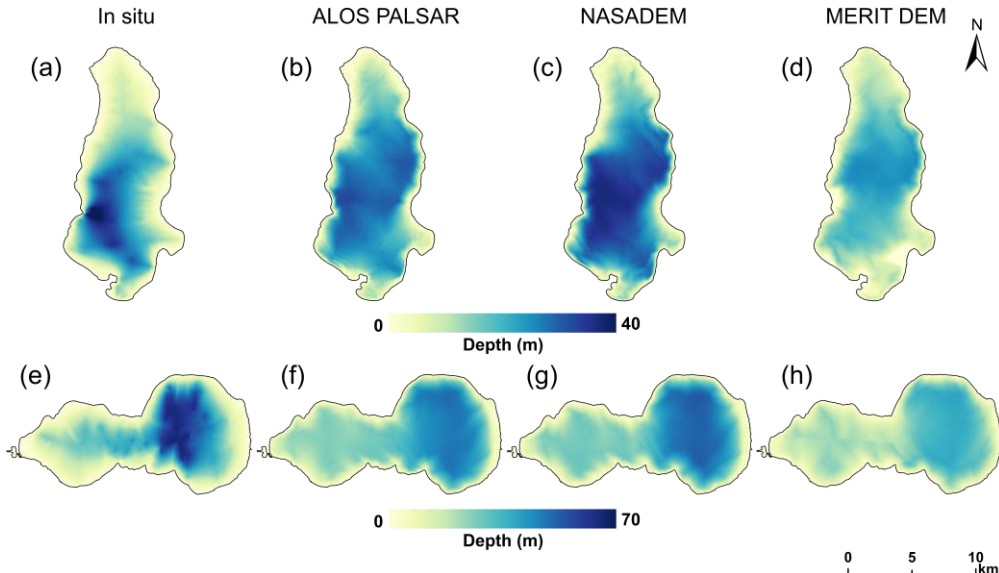


**Figure 10: Comparison of simulated bathymetry maps generated using input DEMs with different spatial resolutions. Bathymetry maps for (a-d) Guomangcuo and (e-h) Longmucuo were derived from in situ data, ALOS PALSAR (12.5 m), NASADEM (30 m), and MERIT DEM (90 m), respectively.**

**Table 4.** Comparison of simulation results using input data with different spatial resolutions.

| Lake name | Input data | Maximum depth (m) | PE | Volume (km³) | PE |
|---|---|---|---|---|---|
| Guomangcuo | ALOS PALSAR (12.5 m) | 33.59 | -14.92% | 2.33 | 32.54% |
| | NASADEM (30 m) | 36.83 | -6.73% | 2.38 | 35.14% |
| | MERIT DEM (90 m) | 25.81 | -34.64% | 1.53 | -13.25% |
| Longmucuo | ALOS PALSAR | 49.96 | -26.0% | 2.98 | 10.49% |
| | NASADEM | 53.45 | -20.84% | 3.08 | 14.12% |
| | MERIT DEM | 39.29 | -41.80% | 2.33 | -13.62% |

### 4.3 Method extension and applicability

To evaluate the applicability of the proposed method, we conducted a case study on Lake Mead in the United States (36.25°N, 114.39°W). As an artificial reservoir, Lake Mead differs significantly from the natural lakes analyzed in this study.

Being a river-type reservoir, its underwater topography exhibits significant spatial heterogeneity across different sections of the river channel. Moreover, unlike many lakes on the Tibetan Plateau, which are characterized by thick sediment layers, sediment deposition in Lake Mead is primarily concentrated within the central river channel (Fig. 11c), resulting in a relatively flat underwater terrain (Rosen and Van Metre, 2010). To account for these characteristics, the sediment correction





module was applied selectively—targeting only the central portion of the profile—during simulation. This tailored approach

enabled the generation of a bathymetry map for Lake Mead (Fig. 11b) that more accurately reflects its distinct geomorphological features.

The simulated underwater elevation map (Fig. 11a) showed an overall consistent pattern and morphology with the in situ bathymetry map (Fig. 11b). The error distribution was predominantly centered around zero, although localized areas of

noticeable overestimation and underestimation were evident (Fig. 11c). Overestimations primarily occurred in the sediment-rich areas, which correspond to the original pre-impoundment riverbed. In these regions, applying the correction formula from Eq. (6) led to an underestimation of the slope, thereby contributing to the overestimated elevations. To quantitatively assess the simulation accuracy, 2,500 validation points were randomly selected across the entire lake, and the results were visualized in a scatter plot (Fig. 11d). The results revealed a strong correlation between in situ and simulated water depths ($r$

$= 0.66$). A significant concentration of validation points was observed at elevations above 300 m, while point dispersion amplified with decreasing elevation, indicating greater uncertainty at deeper sections of the lake. Overall, simulation performance tended to degrade with increasing depth. Despite these challenges, the results demonstrate that the proposed method has promising potential for application in lakes and reservoirs across diverse environmental conditions and geomorphological settings.




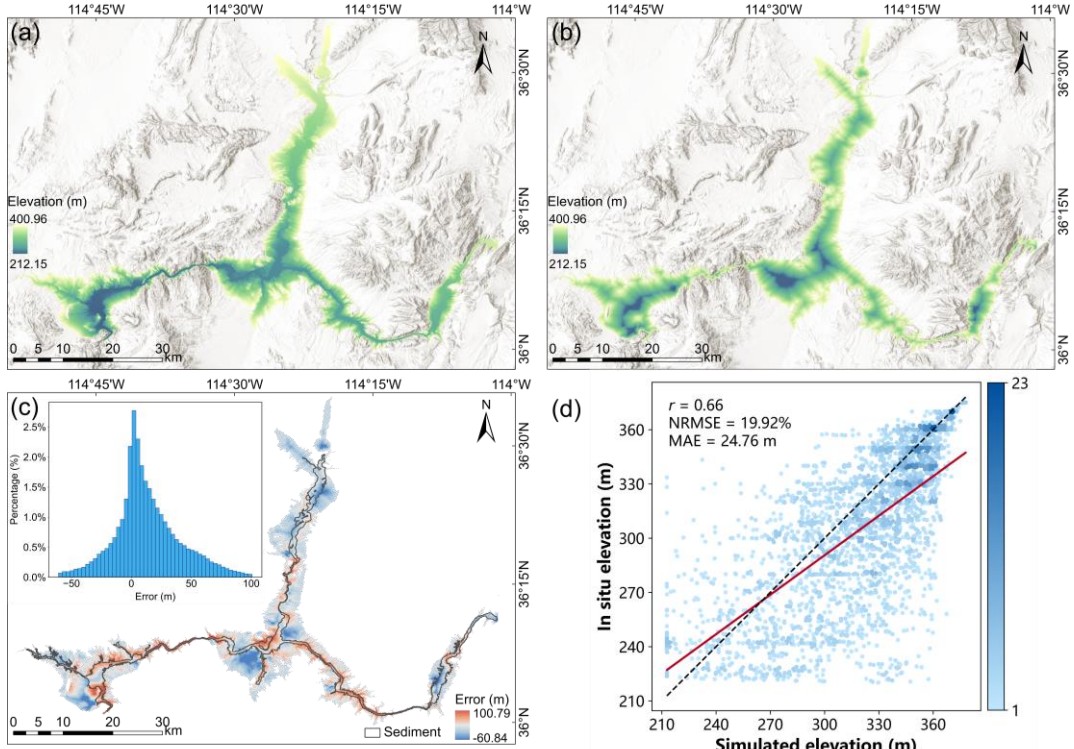

**Figure 11: In situ and simulated bathymetry maps of Lake Mead, along with accuracy evaluation results. (a) In situ bathymetry map. (b) Simulated bathymetry map. (c) Error map between simulated and in situ results, accompanied by a histogram of error distribution. (d) Scatter plot comparing simulated and in situ bathymetric values..**


## 5. Conclusion

This study presents a cost-effective method to predict underwater terrain and lake volume by utilizing DEM data. The proposed approach integrates terrain continuity with lake level recession principles. Applied to 12 lakes on the Tibetan Plateau, the method demonstrates strong performance, achieving an average $r$ of 0.72 and an APE of 23.47% for lake

volume estimates. The simulated bathymetry maps effectively capture underwater terrain variations, providing an effective solution for reconstructing lake depth in areas lacking direct measurements. The primary source of uncertainty in the method stems from the vertical accuracy of the DEM data and error propagation during the simulation process. Since the DEM is the sole input, its quality significantly affects each simulation step. Although NASADEM has shown relatively high vertical accuracy, topography-related errors may still impact results, especially in large-scale applications. Furthermore, simulation

errors tend to accumulate as the model progresses from the shoreline toward the lake center, explaining why depth estimates are generally more accurate near the shoreline than in the deeper central areas of the lake.

To improve accuracy and address current limitations, future efforts should focus on the integration of multi-source datasets. For example, incorporating satellite altimetry data (e.g., ICESat-2), which can partially penetrate water surfaces, could provide valuable constraints for underwater topography estimation. While satellite altimetry alone cannot fully capture underwater elevations, its integration could reduce systematic errors and mitigate the inherent shortcomings of DEM-based methods. Building on this foundation, future work will explore the use of radar-based elevation data to better represent submerged terrain and aim to develop an automated, adaptive framework for lake bathymetry estimation tailored to lake-specific characteristics. Additionally, efforts will be directed toward advancing higher-precision methods for generating lake depth maps, further enhancing the reliability of large-scale hydrological and ecological assessments.

*Code availability*. The codes can be accessed at: https://github.com/WangGugu64/Lake_bathymetry.

*Data availability*. The DEMs used in this study were obtained from publicly available datasets: (1) the ALOS PALSAR DEM provided by JAXA's Earth Observation Research Center (https://www.eorc.jaxa.jp), (2) the NASADEM distributed through NASA's Earthdata portal (https://www.earthdata.nasa.gov), and (3) the MERIT DEM developed by the University of Tokyo (https://hydro.iis.u-tokyo.ac.jp). A bathymetry map of Lake Mead was acquired from the United States Geological Survey (https://www.usgs.gov/).

*Author contributions*. FT: Data curation, Formal analysis, Investigation, Methodology, Validation, Visualization, Writing – original draft. YW: Formal analysis, Writing – review and editing. YJ: Writing – review and editing. XS: Writing – review and editing. SL: Writing – review and editing. YL: Conceptualization, Formal analysis, Funding acquisition, Project administration, Resources, Supervision, Writing – review and editing.

*Financial support*. This work was supported by the National Natural Science Foundation of China (42201349, 42001288), the Chongqing Municipal Science and Technology Bureau (cstc2024ycjh-bgzxm0043), and the Postdoctoral Innovation Talents Support Program of Chongqing (CQBX202322).

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
