# Peer review of "Integrating Topographic Continuity and Lake Recession Dynamics for Improved Bathymetry Mapping from DEMs"

_EGUsphere, 2025_

## Author Comment (AC1)

Dear editors and reviewers,

We sincerely appreciate your constructive comments and suggestions to improve this manuscript. We have revised the manuscript and addressed these comments point by point. We hope that this revised manuscript fulfills the editor's and reviewers' high standards for the *Hydrology and Earth System Sciences*.

The reviewers' comments are shown in black, our responses are highlighted in blue, and the revised text in the manuscript is highlighted in orange.

We look forward to your feedback.

Yours sincerely,

Yao Li

This study is interesting, lake water storage and water depth estimation is important for water resources research, but it is also difficult to get the high accuracy water depth except in situ measurement. This study provided a method to estimate the water depth using the topography similarity, but this method also has a large error comparing with in situ bathymetric data. I suggest that this manuscript need a major revision, and the primary comments as followed.

**Response:** We sincerely appreciate the reviewer's valuable and constructive comments on our manuscript. In response, we have carefully revised the manuscript to address these suggestions. The key revisions include highlighting the methodological innovation, providing details on the maximum water depth, and updating the requested figures and tables. Please find our detailed responses to each comment below.

1. Line 75, Figure 7 shows a large difference between simulate water depth and in situ water depth with a large uncertainty, and the ratio of sediment accumulation is slow, but the authors said that this approach provides a more accurate representation of underwater topography, I had a doubt about it.

**Response 1:** Thank you for this insightful comment. We agree that Fig. 7 shows non-negligible differences between the simulated depths and the in situ data. Although our method seeks to derive reasonably reliable water depths from limited data, uncertainties in the input data and in parts of the workflow can lead to error accumulation.

First, despite the uncertainty in depth estimation, the overall agreement remains acceptable ($r = 0.72$, NRMSE = 19.08%). This performance compares favorably with similar topography-based extrapolation or interpolation approaches.

Second, regarding sediment accumulation, we agree that sedimentation can be slow on annual to decadal timescales. However, the "sediment deposition" component in our framework is intended as a morphological correction that reflects the long-term tendency for deep-water zones to become flatter due to sustained infilling, rather than an attempt to simulate short-term sedimentation rates. In particular, the slope-correction term in Eq. (6) increases from the shoreline toward the lake bottom, indicating that its primary influence is concentrated in deeper areas. Moreover, a synthesis of Tibetan Plateau lake-core records suggests sediment accumulation rates of approximately 0.05–0.06 cm yr$^{-1}$ (Holocene mean) (Yu et al., 2023), implying that sediment infilling is geomorphologically meaningful over centennial to millennial timescales.

In response to your comment, we have revised the manuscript to include the following text: "This approach provides a more effective representation of underwater topography, and supports improved lake volume estimates." (Lines 76–77)

**References**

Yu, L., Cheng, Y., Wang, B., Shi, P., Duan, K., and Dong, Z.: Climate and vegetation codetermine the increased carbon burial rates in Tibetan Plateau lakes during the Holocene, Quaternary Science Reviews, 310, 108118, https://doi.org/10.1016/j.quascirev.2023.108118, 2023

2. Line 75, actually, the similar method had been used in Fang et al. (2023), what the difference between this study and Fang et al. (2023), and point out the real novel of this study.

**Response 2:** Thanks for this comment. The main differences of our method can be summarized as follows:

(1) A new shoreline-slope estimation strategy. When computing terrain slope, we first calculate directional slopes along specific orientations within an eight-neighbor framework. We then rescale the resulting directional-slope values within the buffer zone to match the slope map's value range generated with a conventional 3×3 window. The rationale is to obtain a slope estimate that better represents the shore-to-underwater direction. However, when focusing on a single direction, the slope derived from elevation pixels along that direction may be biased high or low. Rescaling the preliminary directional slopes using the overall slope range in the buffer mitigates this bias, while still retaining stronger directionality than the 3×3 window slope.

(2) A modified implementation of the "lake recession" concept with adaptive drawdown steps. In fact, the strongest commonality between our approach and Fang et al. (2023) is the use of the recession-based idea (Zhu et al., 2019) to represent the underlying process, as noted in Section 2 (Line 142). In our implementation, each drawdown iteration is constrained by the minimum elevation of the "current shoreline" identified at the beginning of that iteration. The iteration continues until all computed elevations in that loop meet this constraint, rather than imposing a fixed water-level drop at each step. This allows the drawdown magnitude to be determined dynamically at each iteration, making the search for newly exposed pixels during water-level lowering more consistent with a natural drawdown process.

(3) A new profile model for underwater elevation. We adopt a quadratic function as the basic profile form and define a hypothetical lowest point based on the ratio of the two bank slopes, establishing the coordinate system around this point. Using the

assumptions that the lowest point has zero slope and its distance to the shoreline can be inferred, we derive closed-form expressions for slope variations on both sides of the lowest point. A key advantage is that these expressions remain valid throughout the iterative drawdown process, even when the water surface has not yet reached the same level across the profile. In the method of Fang et al. (2023), the underwater elevation profile is derived from Zhu et al. (2019), thereby bypassing the need to explicitly locate the lowest point. However, when water levels are not yet consistent during iteration—especially under highly variable terrain, the underlying assumptions are not fully satisfied, which may amplify errors and impose certain limitations.

In summary, our study introduces (1) a slope estimation approach that balances directionality and robustness, (2) an improved recession-based drawdown procedure with adaptive step size, and (3) a new underwater profile model for elevation reconstruction. We have also revised the relevant statements in the Introduction to highlight these novel contributions more clearly.

In response to your comment, we have revised the manuscript to include the following text: "To better capture the representative shore-to-lake gradient, we estimated shore slope using a directional and robust scheme. Directional slopes were first computed along multiple orientations (eight-neighbor directions) and then rescaled within the buffer zone to match the magnitude range of a conventional 3×3-window slope map, thereby preserving directionality while reducing biases associated with single-direction calculations." (Lines 138–142).

"These elevations were estimated using a new profile-based underwater elevation model. This model adopts a quadratic function as the base form and defines an assumed lowest point based on the relative slopes of the two banks. Closed-form expressions then describe slope and elevation variations on both sides of this point, remaining applicable throughout the iterative drawdown." (Lines 148–151).

**References**

Fang, C., Lu, S., Li, M., Wang, Y., Li, X., Tang, H., and Odion Ikhumhen, H.: Lake water storage estimation method based on similar characteristics of above-water and underwater topography, J. Hydrol., 618, 129146, https://doi.org/10.1016/j.jhydrol.2023.129146, 2023.

Zhu, S., Liu, B., Wan, W., Xie, H., Fang, Y., Chen, X., Li, H., Fang, W., Zhang, G., Tao, M., and Hong, Y.: A New Digital Lake Bathymetry Model Using the Step-Wise Water Recession Method to Generate 3D Lake Bathymetric Maps Based on DEMs,

Water, 11, 1151, https://doi.org/10.3390/w11061151, 2019.

3.  Line 105, how did you get the lake boundary? From the results of interpolation? Why did you get the lake boundary from Landsat image, which the time of Landsat is consistent with the measured time of these in situ data.

**Response 3:** Thanks for this comment. As noted at Line 106, we processed the in situ data by identifying points at 0 m depth to delineate the lake boundary. We used this boundary as the spatial extent for calculating lake depth and volume. This is because our accuracy assessment essentially treats the in situ bathymetry as the reference. In contrast, the boundary used in the underwater-terrain reconstruction step is derived from the DEM (Line 159), because we first need to identify the water-covered area represented in the DEM.

In addition, before conducting the experiments, we compared the boundary extracted from the in situ data with the contemporaneous boundary extracted from satellite imagery. The lake-boundary extraction from imagery is briefly described as follows: we used Sentinel-2 data, applied the QA60 mask to remove cloud-contaminated pixels, and calculated NDWI. We then generated a monthly composite using the mean NDWI and applied Otsu thresholding to segment the water body. Here we present two examples (Angzicuo and Guomangcuo). Results show that the boundary derived from in situ data is generally consistent with that extracted from Sentinel-2, and both are larger than the lake boundary in the DEM. As an enhanced version of the SRTM DEM, NASADEM only represents the lake extent around the year 2000. However, lakes on the Tibetan Plateau have expanded continuously over the past three decades (Xu et al., 2024), which explains why the lake boundaries derived from the in situ data and Sentinel-2 are both larger than those from the DEM.

In practice, our workflow first computes and replaces underwater elevations within the DEM-derived lake extent. We then extract the final depth map using the lake boundary corresponding to the target period. Therefore, even when the boundary used to extract the lake depth map is larger than the DEM-derived boundary, the accuracy evaluation remains reliable as long as the spatial extent of the simulated depth map matches that of the in situ bathymetry used for validation. Based on these considerations, we directly adopted the lake boundary derived from the in situ dataset for depth and volume calculation.

[Figure]

**Fig. R1.** Comparison of lake boundaries from different data sources. (a) Lake boundary of Angzicuo. (b) Lake boundary of Guomangcuo. The blue outline denotes the lake extent from the DEM data, the black outline denotes the lake extent from the in situ data, and the red outline denotes the lake extent extracted from Sentinel-2 imagery.

**References**

Xu, F., Zhang, G., Woolway, R. I., et al.: Widespread societal and ecological impacts from projected Tibetan Plateau lake expansion, Nat. Geosci., 17, 516–523, https://doi.org/10.1038/s41561-024-01446-w, 2024.

4. Table 1, suggest to add the measured date, because the water depth of these lakes is changing in recent years.

**Response 4:** Thanks for this suggestion. The measurement date has been added to Table 1. It should be noted that the exact measurement date for Mangcuo is unavailable. However, the lake area of Mangcuo has remained stable in recent years, so these available measurements still provide useful reference information.

Table 1. Overview of the sample lakes on the Tibetan Plateau.

| Lake ID | Lake name | Measurement time | Area (km²) | Average elevation (m) | Depth (m) | |
|---------|-----------|------------------|------------|----------------------|-----------|-----------|
| | | | | | Average | Maximum |
| I | Angzicuo | 2018.09 | 495.42 | 4693 | 9.70 | 18.83 |
| II | Buruocuo | 2013.10 | 92.67 | 5171 | 41.63 | 100.55 |
| III | Dongcuo | 2019.07 | 106.79 | 4397 | 2.06 | 3.99 |

| IV | Guomangcuo | 2019.07 | 113.63 | 4634 | 15.48 | 39.49 |
| V | Laangcuo | 2017.09 | 252.56 | 4571 | 21.90 | 49.19 |
| VI | Longmucuo | 2015.09 | 106.80 | 5009 | 25.27 | 67.52 |
| VII | Mangcuo | - | 19.39 | 4299 | 9.41 | 22.28 |
| VIII | Mapang Yongcuo | 2017.09 | 413.22 | 4585 | 41.62 | 79.45 |
| IX | Ngangla Ringco | 2017.09 | 498.06 | 4715 | 18.26 | 74.94 |
| X | Salt Lake | 2019.11 | 209.90 | 4469 | 13.13 | 32.78 |
| XI | Siling Co | 2014.08 | 2389.11 | 4539 | 22.07 | 52.50 |
| XII | Taro Co | 2012.06 | 487.49 | 4570 | 57.48 | 130.95 |

5. Why did the author select a USA lake, and an artificial reservoir to assess the method's applicability? Maybe the reservoir had a large different with natural lakes, especially for Tibetan Plateau.

**Response 5:** Thanks for this comment. Lake Mead was formed by impoundment behind the Hoover Dam on the Colorado River, with the dam located at the southwestern end of the lake. As noted in the manuscript, it is a typical river-type reservoir. Although our experiments on the Tibetan Plateau include lakes spanning a range of sizes and morphologies, they do not adequately represent this elongated reservoir geometry.

Although Lake Mead is an artificial reservoir, it was created by damming a natural river channel. Therefore, its submerged topography still retains a degree of linkage to the surrounding riverbank terrain. However, as you pointed out, its underwater geomorphology differs substantially from the tectonically formed lakes that dominate the Tibetan Plateau. For this reason, we adjusted part of the computation workflow when applying the method to Lake Mead (Line 470). Our purpose in testing the method in a completely different setting was to explore potential limitations and identify directions for future improvement.

We designed the workflow in a modular manner (see Code availability) so that individual components can be replaced as the method evolves. In this sense, the Lake Mead case should be regarded as an initial step toward broader applications of the proposed method.

6.    Figure 7, whether the authors could redraw this figure with density of these points for different color. Besides, the error of this method for many points is too large, so that I doubt whether this method could provide a more accuracy water depth for water storage estimation or other research. For instance, for a lake, parts of these water depth are overestimation, and parts of these water depth are underestimation, leading to a high accuracy of water storage estimation comparing with in situ bathymetric data, therefore, whether this method is meaningful?

**Response 6:** Thank you for the helpful suggestions. We have revised Figures 7 and 12 to improve readability by visualizing point density with a color scale (rather than plotting all points with a single color), thereby alleviating overplotting and making the distribution of residuals more interpretable. The evaluation still uses 2,500 randomly generated validation points within each lake boundary, and the 1:1 line and regression line are retained for consistency with the original assessment.

Regarding the concern that large errors may undermine the usefulness of the method for water storage estimation. We agree that local overestimation and underestimation exist, especially in lakes with complex bathymetry or weak shoreline constraints. Our results already show that errors tend to increase with depth (e.g., MAE is strongly correlated with lake depth), indicating higher uncertainty in deeper central areas where the correlation between underwater terrain and shoreline features weakens.

This behavior is also consistent with our discussion that simulation errors can accumulate from the shoreline toward the lake center, making nearshore depths generally more reliable than deep-lake estimates.

[Figure]

Figure 7. Scatter plots comparing simulated lake depths with in situ measurements. The dashed line represents the 1:1 line, and the red line represents the linear regression fit.

[Figure]

Figure 12. In situ and simulated bathymetry maps of Lake Mead, along with accuracy evaluation results. (a) In situ bathymetry map. (b) Simulated bathymetry map. (c) Error

map between simulated and in situ results, accompanied by a histogram of error distribution. (d) Scatter plot comparing simulated and in situ bathymetric values.

7.   Table 2, whether the maximum depth for one lake is located in same location between in situ bathymetric data and simulated water depth?

**Response 7:** Thanks for this critical comment. The "maximum depth" is extracted independently as the largest depth value within each bathymetry map. Indeed, our method estimates underwater elevations by propagating shoreline-derived information inward, and uncertainties may accumulate toward the lake center. Given this limitation, we consider it more appropriate to compare the maximum-depth area rather than a single maximum-depth point.

Here, we illustrate this using two representative lakes: Mangcuo, with a relatively fragmented/complex shoreline, and Longmucuo, with a more intact and regular shoreline. The red outline delineates the region where water depths exceed the 95th percentile (i.e., the deepest 5% of pixels). We use this region as a proxy for the "maximum-depth area" in both the in situ and simulated bathymetry maps. As shown, the maximum-depth areas do not perfectly overlap in either lake; however, the deepest zone inferred from the simulated bathymetry still exhibits a meaningful spatial correspondence with the in situ data, indicating that the predicted location of the maximum-depth area can serve as a useful reference.

In response to your concern, we have revised the manuscript to include the following text: "The maximum water depth was derived as the maximum pixel value in the bathymetry map. It should be noted that, in the simulated bathymetry maps, the location of the maximum depth pixel does not necessarily coincide with that in the in situ bathymetry map; nevertheless, the simulated deepest zone remains informative and provides a useful reference (Fig. S3)." (Lines 369–372)

[Figure]

Fig. S3. Bathymetry maps derived from the in situ dataset and from our simulations. (a) In situ bathymetry map of Mangcuo. (b) Simulated bathymetry map of Mangcuo. (c) In situ bathymetry map of Longmucuo. (d) Simulated bathymetry map of Longmucuo.

8. If all simulated points also had a large error, I doubt that the water storage estimation is not meaningful. For instance, Taro Co, the error of many points is large than 50% or 100%, therefore, whether the authors think that the results of water depth will affect the results of water storage estimation.

**Response 8:** Thanks for this comment. Lake water storage is computed from the entire bathymetry map, so both local overestimation and underestimation can affect the storage estimate. We have already highlighted this issue in Section 3.1 using Ngangla Ringco as a representative example (Line 355), where positive and negative depth errors in different parts of the lake may partially offset each other in the integrated volume. We acknowledge that this phenomenon cannot be fully avoided with a DEM-driven approach, and it should be treated as an important factor in future work—especially when interpreting volume agreement in relation to errors at individual locations.

Because our method relies heavily on surrounding topography, localized geomorphic changes can also affect simulation performance. For example, post-formation processes, such as glacial erosion along lake margins, may modify nearshore terrain in ways that are not fully consistent with present-day underwater morphology, potentially leading to

local mismatches.

Despite these limitations, a large proportion of lakes on the Tibetan Plateau are associated with tectonically controlled basins, for which basin-scale geomorphology can still provide meaningful constraints on overall depth patterns. This is also consistent with existing large-scale datasets and empirical approaches: for instance, products such as HydroLAKES estimate lake depth using geomorphic predictors (e.g., slope, lake area) (Messager et al. 2016), and Han et al. (2024) summarized empirical relationships for lake volume based on geological/tectonic settings derived from in situ observations. In this sense, our volume estimation can be viewed as a more explicit and spatially detailed use of lake geometric properties and topographic factors. Therefore, we consider the regional-scale water storage results useful as a general reference, while clearly acknowledging their limitations for local depth accuracy.

**References**

Messager, M. L., Lehner, B., Grill, G., Nedeva, I., and Schmitt, O.: Estimating the volume and age of water stored in global lakes using a geo-statistical approach, Nat. Commun., 7, 13603, https://doi.org/10.1038/ncomms13603, 2016.

Han, X., Zhang, G., Wang, J., Tseng, K.-H., Li, J., Woolway, R. I., Shum, C. K., and Xu, F.: Reconstructing Tibetan Plateau lake bathymetry using ICESat-2 photon-counting laser altimetry, Remote Sens. Environ., 315, 114458, https://doi.org/10.1016/j.rse.2024.114458, 2024.

9.  I suggest that the elevation profiles for each lakes should marked in Figure 1.

**Response 9:** Thanks for this suggestion. We have marked the profiles for each lake in Figure 1, which are also shown in **Figure S2**.

[Figure]

Figure 1. Overview of the 12 sample lakes on the Tibetan Plateau. (a) Distribution of lakes larger than 1 km² across the Tibetan Plateau, with the sample lakes in this study highlighted by red rectangles. (b) Shape characteristics of the 12 sample lakes, with colors indicating differences in mean lake depth. The red line denotes the transect used for validation in Figure 8.

10. Figure 11, the error for Mead lake is also large, many points are underestimation or overestimation. I also think that the accuracy of water depth is much more important than that of water storage estimation.

**Response 10:** We agree that the results for Lake Mead are less stable than those for natural lakes on the Tibetan Plateau. As shown in Figure 11, the overall error is centered around 0, but there are substantial local areas of overestimation and underestimation. Furthermore, the dispersion of the scatter plots increases at deeper water (lower elevations), and the simulation performance decreases with increasing depth. This is related to the characteristics of Lake Mead as a channel-type reservoir, the strong spatial heterogeneity of underwater topography in different river sections, and the fact that sediments are mainly concentrated in the central channel.

As you pointed out, the accuracy of water-depth estimation is crucial and forms the foundation for applications of bathymetry maps. We fully agree with this. However, because water depth is inferred primarily from topographic information, the proposed approach is subject to inherent limitations and uncertainties, and thus still exhibits unavoidable errors in some cases (e.g., in deeper waters or areas with complex geomorphology). To further improve the simulation accuracy of the water-depth maps, we are actively exploring new strategies. For example, we are currently investigating the use of surface water occurrence information to complement a sole topography-driven constraint, which could substantially improve the general applicability and robustness of the approach. We will incorporate these improvements in future work.

---

## Author Comment (AC2)

Dear editors and reviewers,

We sincerely appreciate your constructive comments and suggestions to improve this manuscript. We have revised the manuscript and addressed these comments point by point. We hope that this revised manuscript fulfills the editor's and reviewers' high standards for the *Hydrology and Earth System Sciences*.

The reviewers' comments are shown in black, our responses are highlighted in blue, and the revised text in the manuscript is highlighted in orange.

We look forward to your feedback.

Yours sincerely,

Yao Li

This manuscript presents a promising and innovative framework for reconstructing lake bathymetry by leveraging topographic continuity and widely available Digital Elevation Models (DEMs). The study addresses a critical need for cost-effective alternatives to traditional surveys, and the validation effort involving 12 lakes on the Tibetan Plateau represents a substantial and valuable contribution to the field. While the work is well-structured and tackles a clearly defined problem, the manuscript would benefit from the following refinements to further strengthen its theoretical grounding and clarify its methodological contributions.

**Response:** We sincerely appreciate the reviewer's positive feedback. We have carefully revised and improved the manuscript based on your comments. The key revisions include strengthening the theoretical grounding and adding a sensitivity experiment on the buffer distance. Please find our detailed responses to each comment below.

1. The manuscript's conceptual framing could be significantly strengthened by realigning the "lake recession" terminology with the well-documented hydrological expansion of lakes on the Tibetan Plateau.

**Response 1:** Thank you for this important suggestion. We agree that the term "lake recession" in the manuscript could be misinterpreted as describing an observed hydrological trend, whereas many lakes on the Tibetan Plateau have shown well-documented expansion in recent decades. However, the concept of "lake recession" was not originally proposed in our study, and we therefore retain this terminology. To avoid confusion, we have added further clarification in the manuscript to explicitly state that "lake recession" is used here only as a physical simulation within the modeling procedure and does not imply a real, long-term lake-level recession: "lake level recession (used here solely as a physical simulation within the reconstruction procedure, rather than implying an observed lake-level trend)" (Lines 76–77)

2. Reframing the method's success as leveraging "historical exposure" captured in older DEMs (e.g., SRTM 2000) prior to inundation would better articulate the physical mechanism driving the accurate results.

**Response 2:** Thanks for this insightful suggestion. We agree that the performance of the proposed method is fundamentally linked to the physical information preserved in historical DEMs acquired before lake inundation.

Specifically, NASADEM captured large portions of lake margins and shallow basins before the widespread hydrological expansion observed in recent decades on the

Tibetan Plateau. These historically exposed terrains provide critical geomorphic constraints that can be leveraged to infer present-day submerged topography.

In response to your suggestion, we have revised the manuscript to include the following text: "Consequently, the DEM preserves geomorphic information from historically exposed shorelines and shallow lake margins, providing critical physical constraints for reconstructing present-day underwater topography and enabling more comprehensive lake depth estimation." (Lines 87–91)

3. In order to enhance the study's robustness, it would be beneficial to elaborate on the rationale for selecting the 12 validation lakes. For instance, classifying these lakes by geomorphological origin (e.g., tectonic, glacial) and discussing the algorithm's consistency across these types would greatly increase the paper's utility for the broader research community.

**Response 3:** Thanks for this helpful suggestion. In the revised manuscript, we have expanded the justification for selecting the 12 validation lakes and clarified their representativeness. Specifically, these lakes were chosen for the following reasons:

(1) High-quality in situ bathymetry datasets (echo-sounder surveys) are available for them from the National Tibetan Plateau/Third Pole Environment Data Center, enabling point-wise and volume-based validation.

(2) They are evenly distributed across the Tibetan Plateau and span a wide range of lake characteristics (area, elevation, depth, shoreline complexity), which is essential for testing model robustness.

Following your suggestion, we attempted to classify the lakes used in this study in a more detailed manner based on their geomorphological origin. First, the formation of all 12 lakes is closely related to geological processes on the Tibetan Plateau, and Buruo Co has been identified as a proglacial lake (Xu et al., 2019). Building on this, we further explored differences in lake types from a structural–tectonic perspective. For lakes whose long axes are approximately parallel to major fault trends (i.e., (c) Dongcuo, (f) Longmucuo, (h) Mapang Yongcuo, and (k) Siling Co), all except Dongcuo showed good performance in bathymetry estimation ($r = 0.86$, 0.78, and 0.83). This result may indirectly suggest that our method performs better when underwater morphology is strongly coupled with surrounding topography.

Nevertheless, given the limited sample size in this study, drawing definitive conclusions about the relationship between our method's performance and deeper tectonic controls could undermine the rigor of the manuscript. Therefore, we provide Fig. S4 in the

Supplementary for readers' reference rather than discussing it in detail in the main text. In future work, we will carefully follow this suggestion and, using a substantially larger sample set, investigate how geomorphological origin influences the performance of topography-based bathymetric methods to derive more robust and broadly applicable conclusions.

[Figure]

**Figure S4.** Spatial relationships between major faults (red lines) and the long-axis orientations of the 12 validation lakes. Fault data are derived from Gao et al. (2023).

**References**

Xu, T., Zhu, L. P., Lü, X. M., Ma, Q. F., Wang, J. B., Ju, J. T., and Huang, L.: Mid- to late-Holocene paleoenvironmental changes and glacier fluctuations reconstructed from the sediments of proglacial lake Buruo Co, northern Tibetan Plateau, *Palaeogeogr. Palaeoclimatol. Palaeoecol.*, 517, 74–85, https://doi.org/10.1016/j.palaeo.2018.12.023, 2019.

Gao, Z.: Seismotectonic map and seismic hazard zonation map of Pan-Third Pole region (1960–2021), National Tibetan Plateau / Third Pole Environment Data Center, https://doi.org/10.11888/SolidEar.tpdc.300783, 2023.

4. It is suggested to include a sensitivity analysis regarding the width of the "dynamic exposed area" used for slope calculation in Discussions.

**Response 4:** Thank you for this helpful suggestion. In our method, the shoreline slope is estimated from the buffer area (i.e., the "dynamic exposed area") around the lake boundary. Specifically, we adopt a multi-level buffering strategy with a maximum buffer of 600 m and nested buffers at 100 m intervals; a terrain transition point adaptively determines the final buffer extent, and a default buffer of 600 m is used when no transition point is detected.

To evaluate the robustness of the modeled bathymetry to this parameter, we performed a sensitivity analysis by varying the maximum buffer width. We have revised Section 4.2 to discuss the impact of different buffer size: "We further assessed how bathymetry accuracy responds to the distance of the dynamic exposed area used to calculate shoreline slope by testing maximum buffer widths of 300, 600, and 900 m (Fig. 11). The median NRMSEs are similar across the three settings (16.65%, 18.00%, and 18.14%, respectively), indicating that overall performance is not strongly sensitive to buffer width within the tested range. However, error dispersion increases with buffer size, with the 900 m setting exhibiting the largest interquartile range (IQR = 7.72) and a more pronounced upper tail, suggesting that overly wide buffers may incorporate broader-scale topographic signals unrelated to the representative nearshore slope (e.g., terraces or distant hillslopes), thereby degrading performance for some lakes. In contrast, the 600 m setting yields the smallest IQR (6.19) and the most consistent results across lakes; it is therefore adopted as the default in this study. Notably, because our workflow determines an adaptive buffer extent using the multi-level buffering scheme described in Section 2.2.1, the specified value represents the maximum buffer distance and is not necessarily reached for all lakes. In some cases, the optimal buffer is identified at a smaller distance, so the maximum value is not applied." (Lines 473–485).

[Figure]

Figure 11. Boxplots of NRMSE (%) from buffer distance sensitivity experiments (300, 600, and 900 m). Box ranges represent the upper and lower quartiles, and whiskers extend to 1.5 times the interquartile range.

5. For the proposed Method, it would be better to suggest a recommended threshold for this exposed zone would provide valuable guidance for users applying this method to lakes with varying bank steepness.

**Response 5:** Thank you for this suggestion. In our workflow, the "exposed zone" (dynamic exposed area) used for shoreline-slope estimation is defined through a multi-level buffering strategy (100 m intervals) with an upper bound, and the final buffer extent is adaptively determined by a terrain transition point; if no transition point is detected, a default maximum buffer is used.

Based on the sensitivity experiment (maximum buffer = 300/600/900 m), the median NRMSE varies slightly across settings (16.65%, 18.00%, and 18.14%), indicating limited sensitivity within this range. Notably, the 600 m setting yields the smallest interquartile range (IQR = 6.19), reflecting the most consistent performance across lakes. In contrast, a larger maximum buffer (900 m) increases dispersion (IQR = 7.72) and may introduce broader-scale terrain signals unrelated to the representative nearshore slope.

Therefore, we recommend setting the maximum exposed-zone width to ~600 m (≈15–20 pixels for 30 m DEMs) as a default upper bound, together with the built-in adaptive selection. For lakes with steeper banks, we further suggest using a smaller upper bound (e.g., 300–600 m) to avoid mixing distant hillslopes/terraces into the slope fit; for gentle banks, 600 m remains appropriate, and a larger upper bound (e.g., 900 m) should only

be used when the nearshore terrain is very flat or when the transition point cannot be identified robustly. As a practical criterion for "bank steepness", users may adopt the mean shoreline slope threshold of ~5° (already used in our parameterization).

We have revised Section 4.2, and users can also refer to the performance of different buffer distances to select an appropriate delineation threshold.

6. The error analysis would be more impactful if it moved beyond listing discrepancies to offering a geomorphological diagnosis of the results. Explicitly linking performance variations (e.g., in Dongcuo and Ngangla Ringco) to factors such as signal-to-noise ratios or structural decoupling would add significant depth to the findings.

**Response 6:** Thank you for this insightful suggestion. We agree that a geomorphological diagnosis provides more actionable interpretation than a simple error listing. Accordingly, we revised Section 3.1 to explicitly connect performance variations to the signal-to-noise ratio of shoreline-derived slope information, and potential structural decoupling between shoreline topography and lakebed morphology:

"To quantitatively evaluate the accuracy of the simulated bathymetry, we randomly generated 2,500 validation points within each lake boundary and compared simulated depths against in situ measurements. As shown in Fig. 7, the simulated depths exhibit good agreement with observations, with an average $r$ value of 0.72 and an average NRMSE value of 19.09%. Because the method assumes that nearshore topography contains informative signatures of underwater slope structure, its performance depends on the signal-to-noise ratio (SNR) of shoreline-derived slope information and the degree of structural coupling between shoreline morphology and lakebed geometry within a basin.

According to Table 2, the method underestimated maximum water depth for several lakes, including Angzicuo, Buruocuo, Longmucuo, Ngangla Ringco, and Taro Co. Among them, Ngangla Ringco (Fig. 7i) shows the weakest agreement. Inspection of the three-dimensional bathymetry (Fig. 6i) reveals an abrupt deepening in the southern part of the lake, indicating a localized bathymetric anomaly and partial shoreline–lakebed decoupling. Such features are difficult to infer from shoreline terrain alone, posing a challenge for approaches that rely primarily on nearshore topographic constraints. In contrast, simulated depths were overestimated for Dongcuo and Mapang Yongcuo. In particular, Dongcuo, with a surface area of 106.80 km² and a maximum depth of 3.99 m, exhibits relatively poor simulation performance despite its shallow

depth. This likely reflects the limited depth range and the heightened influence of measurement noise and subtle topographic gradients, which can reduce the robustness of shoreline-derived slope signals and amplify relative errors.

Across all lakes, MAE is strongly correlated with lake depth ($r = 0.94$). For deep lakes such as Buruocuo and Taro Co, where maximum water depths exceed 100 m, MAE values reached 17.33 m and 21.15 m, respectively. This pattern suggests that shoreline–lakebed coupling tends to weaken with increasing depth, consistent with a reduced ability of shoreline-derived constraints to represent deep-basin morphology. Nevertheless, the model maintains acceptable accuracy across a wide range of lake sizes, depths, and morphologies, demonstrating its general applicability for regional-scale bathymetric estimation." (Lines 337–359).

7. The discussion in Section 4.2 regarding the performance of NASADEM versus ALOS PALSAR offers an opportunity for deeper insight. Highlighting the temporal advantage of the older NASADEM (acquired during low stands) rather than focusing solely on spatial resolution would provide a compelling explanation for its superior performance.

**Response 7:** Thanks for your insightful suggestion. We agree that the performance difference between NASADEM and ALOS PALSAR should not be interpreted solely in terms of spatial resolution. We have revised Section 4.2 to explicitly emphasize the temporal (acquisition-epoch) advantage of NASADEM. Specifically, NASADEM (improved SRTM DEM) was acquired earlier and can preserve more exposed nearshore terrain under relatively lower lake stands, providing more reliable shoreline gradients for our recession-based bathymetry inference. Because our method relies on the water mask and shoreline slope derived from the input DEM, a DEM acquired during lower water levels can better constrain the shore-to-lake transition and reduce error propagation toward the lake center. This temporal factor offers a compelling explanation for why NASADEM outperforms the higher-resolution ALOS PALSAR within our framework.

In response to your suggestion, we revised the manuscript to include the following text:

"Beyond spatial resolution, the DEM acquisition time is also critical for our method. Because the algorithm infers underwater elevations from shoreline gradients and a DEM-derived water mask, DEMs acquired at lower lake levels can preserve more exposed nearshore topography. This additional geomorphic information strengthens constraints on shoreline-slope estimation and the subsequent recession simulation. In

this regard, the earlier acquisition time of NASADEM may partly explain its better performance compared to ALOS PALSAR, despite the latter's finer spatial resolution." (Lines 444–448).

8. Abstract: A minor adjustment to punctuation in the phrase "Bathymetry data and lake volume two key physical parameters" is recommended.

**Response 8:** Thank you for your suggestion. The punctuation has been corrected as recommended: "Bathymetry data and lake volume, two key physical parameters of lakes" (Line 25).

9. Section 1: To correct the typo in the header "Introdution".

**Response 9:** Thanks. We have corrected this typo.